# Extending Confidence-Based Text2Cypher with Grammar and Schema Aware Filtering

Makbule Gulcin Ozsoy[†]

[1]*Neo4j, London, UK*

## Abstract

Large language models (LLMs) allow users to query databases using natural language by translating questions into executable queries. Despite strong progress on tasks such as Text2SQL, Text2SPARQL, and Text2Cypher, most existing methods focus on better prompting, fine-tuning, or iterative refinement. However, they often do not explicitly enforce structural constraints, such as syntactic validity and schema consistency. This can reduce reliability, since generated queries must satisfy both syntax rules and database schema constraints to be executable. In this work, we study how structured constraints can be used in test-time inference for Text2Cypher. We focus on post-generation validation to improve query correctness. We extend a confidence-based inference framework with a sequential filtering process that combines confidence scoring, grammar validation, and schema constraints before final aggregation. This lets us analyze how different constraint types affect generated queries. Our experiments with two instruction-tuned models show that grammar-based filtering improves syntactic validity. Schema-aware filtering further improves execution quality by enforcing consistency with the database structure. However, stronger filtering also increases the number of empty predictions and reduces execution coverage. Overall, we show that adding simple structural checks at test time improves the reliability of Text2Cypher generation, and we provide a clearer view of how syntax and schema constraints contribute differently.

## Keywords

Text2Cypher, Confidence-Based Inference, Grammar and Schema-Aware Filtering, Test-Time Strategies

## 1. Introduction

Databases play an important role in modern knowledge systems by enabling structured storage and efficient querying of data. In order to access these systems, users rely on formal query languages such as SQL for relational databases, SPARQL for RDF graphs, and Cypher for graph databases. Recently, large language models (LLMs) have enabled natural language interfaces to databases, where user questions are translated into executable queries.

This problem has been widely studied in tasks such as Text2SQL, Text2SPARQL, and Text2Cypher. Many existing methods focus on prompt engineering, fine-tuning, or iterative refinement pipelines [1, 2, 3, 4, 5, 6]. However, these approaches largely focus on model training or prompting, while the role of structured constraints during test-time inference remains underexplored. Recent studies show that self-consistency, where multiple candidate queries are generated and aggregated, can improve performance [7]. Confidence-based filtering further improves this process by removing low-quality candidates before aggregation [8], and has been adapted to Text2Cypher in prior work [9]. However, these methods do not explicitly enforce structural constraints, such as syntactic validity and schema consistency. This is particularly important in tasks like Text2Cypher, where queries must follow strict syntax rules and the database schema [10]. Otherwise, they may be invalid or fail to execute.

Grammar-constrained decoding addresses this by restricting generation to valid sequences [10, 11, 12, 13]. Most methods apply these constraints during decoding, while their use as a post-generation filtering step together with confidence-based aggregation remains underexplored. Schema information is also essential for correct query generation, as it links natural language to database elements such as tables, columns, nodes, and relationships [14, 15]. Many works show that schema information improves query generation, especially in Text2SQL and Text2Cypher. However, most schema-aware methods integrate

*GenAIK-NORA: The Joint Workshop on Generative AI and Knowledge Graphs and KNOwledge GRaphs & Agentic Systems Interplay, 2026*

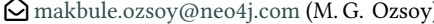 makbule.ozsoy@neo4j.com (M. G. Ozsoy)

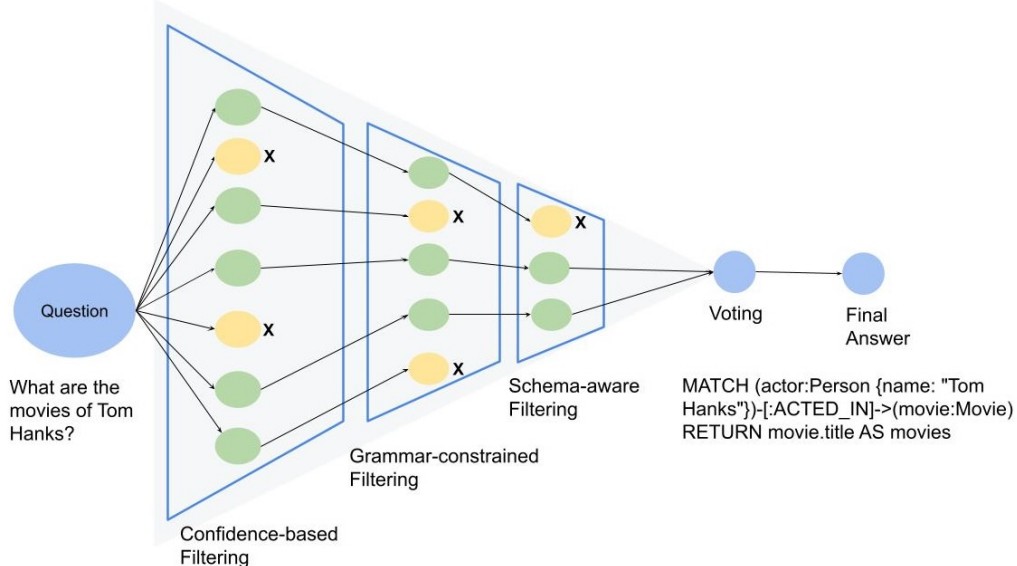

**Figure 1:** Overview of the filtering pipeline, where confidence-, grammar-, and schema-based steps are applied sequentially to remove invalid or low-quality queries before aggregation.

schema information in the input prompt or during decoding, while using it as a post-generation filtering step remains less explored.

In this work, we study how structured constraints can be incorporated into test-time inference for Text2Cypher through a sequential filtering process. Building on a confidence-based inference framework [9], we introduce grammar- and schema-aware filtering of generated query candidates. Starting from a self-consistency framework, we apply a sequence of filters, such as formal grammar validation, and schema constraints, before aggregation. This sequential process progressively removes invalid or inconsistent queries (Figure 1). This allows us to analyze how different constraint types—confidence, syntax, and schema—affect different types of errors in generated queries. Our results show that grammar-based filtering improves syntactic correctness, while schema-aware filtering improves answer quality. However, these gains come with a trade-off, as more aggressive filtering can reduce execution coverage due to empty predictions.

Our main contributions are as follows:

- We study structured constraints in test-time inference for Text2Cypher. In particular, we look at how confidence filtering, grammar validation, and schema constraints work together in a sequential filtering process.
- We extend a confidence-based inference method by adding grammar- and schema-based filtering steps before aggregation. This allows us to test how different types of constraints affect query generation in a clear and controlled way.
- We evaluate our approach in different inference settings, different sampling diversity levels, and two instruction-tuned models. We find that grammar-based filtering improves syntactic validity, while schema-aware filtering improves execution-based answer quality. However, both methods reduce execution coverage because they can remove all candidates in some cases.

The rest of the paper is organized as follows. Section 2 reviews related work. Section 3 describes our proposed method. Section 4 presents the experimental setup and results. Section 5 and Section 6 presents limitations and concludes the paper.

## 2. Related Work

Mapping natural language to formal query languages has been widely studied in semantic parsing [16, 17]. With the rise of large language models (LLMs), significant progress has been made in tasks

such as Text2SQL, Text2SPARQL, and Text2Cypher. Most existing approaches focus on improving model inputs or training strategies, such as prompt engineering, fine-tuning, and iterative refinement pipelines [1, 2, 3, 4, 5, 6]. In contrast, test-time inference strategies have received less attention.

Recent work explores test-time methods such as self-consistency, where multiple outputs are generated and aggregated to improve robustness [7]. This idea has been extended in different ways, including prompt diversification, ensemble methods, and confidence-based filtering [18, 19, 20, 8, 21]. Confidence-based filtering removes low-quality candidates before aggregation [8] and has been adapted to Text2Cypher in a prior work [9]. However, these approaches do not explicitly enforce structural constraints required for executable queries. This is important in tasks such as Text2Cypher, where generated queries must follow strict syntax rules and the database schema [10]. If these constraints are not satisfied, queries may become invalid or not executable.

Grammar-constrained decoding addresses syntactic correctness by restricting generation to valid sequences defined by formal grammars [11, 12, 22, 13, 10]. These methods use context-free grammars, finite-state automata, regular expressions or templates to ensure valid outputs. Previous work shows that grammar constraints improve syntactic validity and parsing accuracy [13, 10]. However, they are mainly applied during decoding, and their use as a post-generation filtering step remains less explored.

Schema information is also essential for correct query generation, as it links natural language to database structures such as tables, columns, nodes, and relationships [14, 15]. Many works show that schema information improves query generation, especially in Text2SQL and Text2Cypher. For these tasks, various methods have been proposed, such as string matching, learning-based models, and more recently LLM-based approaches using prompting, fine-tuning, or agent-based methods [23, 24, 25, 26, 1, 27, 28, 14, 29, 30]. Most schema-aware methods integrate schema information in the input prompt or during decoding. However, using it as a post-generation filtering signal has been less studied.

Overall, prior work has studied confidence-based inference, grammar-constrained decoding, and schema-aware generation mostly in isolation. To the best of our knowledge, combining confidence-based inference with post-generation grammar- and schema-aware filtering has not been systematically studied for Text2Cypher. In this work, we bring these ideas together and show that combining them improves both structural validity and answer quality.

## 3. Methodology

In the Text2Cypher task, the goal is to generate an executable database query from a natural language question. A correct query must satisfy syntactic validity, schema consistency, and produce the correct execution result [9]. We build on prior work on confidence-based inference for Text2Cypher [9], where multiple candidate queries are generated and aggregated based on model confidence. While this improves robustness, it does not explicitly enforce syntactic or schema-level constraints.

In this work, we extend this framework by introducing additional filtering stages before aggregation. Specifically, generated candidates are first filtered using confidence scores, and then refined using grammar- and schema-based constraints. This sequential filtering improves structural validity while preserving high-quality candidates. As illustrated in Figure 1, the process progressively removes invalid or inconsistent queries before final aggregation.

**Confidence-Based Filtering for Text2Cypher**    We adopt the confidence-based inference framework for Text2Cypher proposed in prior work [9], based on Deep Think with Confidence (DeepConf) [8]. The approach generates multiple candidate queries (traces) and aggregates them based on model confidence.

We evaluate two inference modes following DeepConf [8]. In **offline mode**, a fixed set of candidate queries is first generated, then low-confidence candidates are filtered and the remaining outputs are aggregated using voting. In **online mode**, low-confidence generations are terminated early during decoding to reduce computation, and the remaining valid traces are aggregated in a similar manner.

Following previous confidence-aware Text2Cypher work [9], we use an instruction-tuned model with two sampling diversity settings: moderately-diverse (`temperature=0.9, top_p=0.99, top_k=60`)

and very-diverse (`temperature=1.2, top_p=0.999, top_k=80`). While confidence-based filtering improves performance, it does not enforce syntactic correctness or schema consistency. We address this limitation by adding grammar- and schema-based filtering stages.

**Grammar-Based Filtering for Text2Cypher**    In order to ensure syntactic correctness of generated queries, we introduce grammar-constrained filtering within the confidence-based inference framework. Given a set of candidate Cypher queries, we apply grammar-based validation after confidence-based filtering and before aggregation. Queries that do not satisfy the required syntactic constraints are removed from the candidate pool.

We consider two types of grammar-based filtering: (i) **Naive approach:** The naive method applies a set of lightweight checks using regular expressions. It verifies that brackets and parentheses are properly matched, prevents invalid repetitions of clause keywords (e.g. RETURN RETURN), and ensures that the query contains at least one valid Cypher clause such as MATCH, CREATE, RETURN, or WITH. These checks provide a fast and simple way to remove clearly malformed queries. (ii) **Formal grammar validation:** The formal approach uses an ANTLR4 [31] implementation of the OpenCypher grammar [1]. Each candidate query is parsed using the generated lexer and parser, and queries with syntax errors are rejected. Compared to the naive approach, this method enforces stricter and more complete syntactic validation at the grammar level. These two approaches allow us to study the effect of different levels of syntactic strictness on performance. Both validators are applied before the aggregation step.

**Schema-Aware Filtering for Text2Cypher**    In order to further improve answer quality, we introduce schema-aware filtering on top of grammar-based filtering. This step checks whether generated queries are consistent with the database schema, with a particular focus on relationship directionality.

The schema is either provided with the dataset or extracted from the database, e.g., in the form `(:SourceLabel)-[:REL_TYPE]->(:TargetLabel)`. Based on this, we validate whether each relationship in a generated query follows the correct direction. If a relationship type appears in the schema but is used in the opposite direction in a generated query, the query is considered invalid and is removed. This check is applied after grammar-based filtering and before the aggregation step.

## 4. Experiments

### 4.1. Experimental Setup and Evaluation Metrics

For our experiments, we follow the setup from prior work on confidence-based filtering for Text2Cypher [9]. We use a subset of the publicly available Text2Cypher dataset [3], consisting of 789 test samples. These samples are associated with three databases, namely "recommendations", "companies", and "neoflix", which were identified in prior work [9] as more challenging than the remaining instances. We use the same instruction-tuned model as in prior work [9], namely "google/gemma-2-9b-it"[2], referred to as Gemma2 in the rest of the paper. We also use the "Qwen/Qwen2.5-7B-Instruct"[3] model for comparison, referred to as Qwen2.5. For inference, we use vLLM [32]. We also apply a simple post-processing step to remove unwanted text, such as redundant "cypher:" prefixes.

For evaluation, we employ two procedures: (i) **translation-based (lexical) evaluation**, which compares generated queries with ground-truth queries at the text level, and (ii) **execution-based evaluation**, which executes both queries on the target database and compares their outputs. All metrics are computed using the Hugging Face Evaluate library [4]. We report ROUGE-L and error statistics as the primary evaluation metrics. This setup ensures comparability with prior work.

---

[1] https://github.com/antlr/grammars-v4/tree/master/cypher
[2] https://huggingface.co/google/gemma-2-9b-it
[3] https://huggingface.co/Qwen/Qwen2.5-7B-Instruct
[4] https://huggingface.co/evaluate-metric

**Table 1**

Translation (lexical) and execution metrics for Gemma2 across inference modes and filtering strategies, showing incremental improvements from confidence-, grammar-, and schema-based filtering. Performance improvements over the base model are indicated as superscripts, scaled to percentage points for ROUGE-L scores.

| Diversity Level | Inference Mode | Filtering | Variant | ROUGE-L (Lexical) | ROUGE-L (Exec.) | Exec. Succ. Ratio (%) |
|---|---|---|---|---|---|---|
| Moderately-Diversed | Base | — | — | 0.6818 | 0.1989 | 77.6 |
| | Online | Confidence | — | **0.7160**$^{+3.42}$ | 0.2402$^{+4.13}$ | **85.4**$^{+7.8}$ |
| | | + Grammar | Naive | **0.7125**$^{+3.07}$ | 0.2402$^{+4.13}$ | **85.4**$^{+7.8}$ |
| | | | ANTLR | 0.7034$^{+2.16}$ | 0.2351$^{+3.62}$ | **86.7**$^{+9.1}$ |
| | | + Schema | Naive+Schema | 0.7004$^{+1.86}$ | **0.3025**$^{+10.36}$ | 83.6$^{+6.0}$ |
| | | | ANTLR + Schema | 0.6884$^{+0.66}$ | **0.2984**$^{+10.05}$ | 84.0$^{+6.4}$ |
| | Offline | Confidence | — | **0.7081**$^{+2.63}$ | 0.2366$^{+3.77}$ | 81.7$^{+4.1}$ |
| | | + Grammar | Naive | **0.7100**$^{+2.82}$ | 0.2468$^{+4.79}$ | **84.2**$^{+6.6}$ |
| | | | ANTLR | 0.7022$^{+2.04}$ | 0.2419$^{+4.30}$ | **86.7**$^{+9.1}$ |
| | | + Schema | Naive + Schema | 0.6980$^{+1.62}$ | **0.2945**$^{+9.56}$ | 79.9$^{+2.3}$ |
| | | | ANTLR + Schema | 0.6823$^{+0.05}$ | **0.2903**$^{+9.14}$ | 83.1$^{+5.5}$ |
| Very-Diversed | Base | — | — | 0.6284 | 0.1680 | 63.8 |
| | Online | Confidence | — | 0.7099$^{+8.15}$ | 0.2386$^{+7.06}$ | 83.7$^{+19.9}$ |
| | | + Grammar | Naive | **0.7155**$^{+8.71}$ | 0.2409$^{+7.29}$ | 83.0$^{+19.2}$ |
| | | | ANTLR | **0.7133**$^{+8.49}$ | 0.2500$^{+8.20}$ | **86.7**$^{+22.9}$ |
| | | + Schema | Naive + Schema | 0.7021$^{+7.37}$ | **0.2966**$^{+12.86}$ | 80.8$^{+17.0}$ |
| | | | ANTLR + Schema | 0.6973$^{+6.89}$ | **0.3025**$^{+13.45}$ | **85.1**$^{+21.3}$ |
| | Offline | Confidence | — | 0.6950$^{+6.66}$ | 0.2216$^{+5.36}$ | 75.1$^{+11.3}$ |
| | | + Grammar | Naive | **0.7130**$^{+8.46}$ | 0.2345$^{+6.65}$ | **82.2**$^{+18.4}$ |
| | | | ANTLR | **0.7009**$^{+8.28}$ | 0.2377$^{+6.97}$ | **84.1**$^{+20.3}$ |
| | | + Schema | Naive + Schema | 0.6961$^{+6.77}$ | **0.2839**$^{+11.59}$ | 79.3$^{+15.5}$ |
| | | | ANTLR + Schema | 0.6901$^{+6.17}$ | **0.2880**$^{+12.00}$ | 82.1$^{+18.3}$ |

## 4.2. Evaluation Results

We evaluate the effectiveness of the proposed confidence-, grammar-, and schema-based filtering framework for Text2Cypher from multiple perspectives. First, we analyze the impact of grammar-based filtering using both naive and formal grammar validation. Second, we study the effect of schema-aware filtering on top of the grammar-based filtering stage. We evaluate these components under different inference settings, including offline and online modes of DeepConf [8], as well as a base setting without confidence-based aggregation. In addition, we compare results across different sampling diversity levels.

**Effect of grammar-based filtering** We first analyze the effect of grammar-based filtering applied on top of confidence-based inference. Consistent with prior work [9], confidence-based filtering already provides a large improvement in performance (Table 1). Grammar-based filtering further refines the candidate set by removing invalid queries before aggregation.

Among the filtering strategies, formal grammar validation (ANTLR-based) is more effective than the naive rule-based approach, as it removes a larger number of syntactically invalid queries. As a result, ANTLR-based filtering achieves the highest execution success rates across both diversity settings (Table 1). In contrast, the naive approach is less restrictive, it preserves more candidates, but provides weaker improvements in syntactic correctness. However, stricter filtering introduces a trade-off. In some cases, all candidates are removed, resulting in empty outputs (Table 2). For example, ANTLR-based filtering produces up to 24 empty cases in the moderately-diversed setting and 18 cases in the very-diversed setting. Interestingly, even the base model produces some empty outputs, likely due

**Table 2**

Execution error analysis for Gemma2 across filtering strategies, showing trade-offs between error reduction, syntactic correctness, and empty predictions.

| Diversity Level | Inference Mode | Filtering | Variant | Succ.(%) | Run.↓ Err | Syn. ↓ Err | Empty↓ |
|---|---|---|---|---|---|---|---|
| Moderately-Diversed | Base | — | — | 77.6 | 29 | 136 | 11 |
| | Online | Confidence | — | 85.4 | 34 | 80 | 1 |
| | | + Grammar | Naive | 85.4 | 32 | 83 | 0 |
| | | | ANTLR | 86.7 | 38 | 43 | 24 |
| | | + Schema | Naive+Schema | 83.6 | 35 | 81 | 13 |
| | | | ANTLR+Schema | 84.0 | 35 | 55 | 36 |
| | Offline | Confidence | — | 81.7 | 35 | 108 | 1 |
| | | + Grammar | Naive | 84.2 | 28 | 96 | 0 |
| | | | ANTLR | 86.7 | 34 | 51 | 20 |
| | | + Schema | Naive+Schema | 79.9 | 36 | 108 | 14 |
| | | | ANTLR+Schema | 83.1 | 36 | 62 | 35 |
| Very-Diversed | Base | — | — | 63.8 | 29 | 206 | 50 |
| | Online | Confidence | — | 83.7 | 28 | 96 | 4 |
| | | + Grammar | Naive | 83.0 | 35 | 99 | 0 |
| | | | ANTLR | 86.7 | 33 | 54 | 18 |
| | | + Schema | Naive+Schema | 80.8 | 38 | 106 | 7 |
| | | | ANTLR+Schema | 85.1 | 36 | 54 | 27 |
| | Offline | Confidence | — | 75.1 | 36 | 156 | 4 |
| | | + Grammar | Naive | 82.2 | 31 | 109 | 0 |
| | | | ANTLR | 84.1 | 33 | 74 | 18 |
| | | + Schema | Naive+Schema | 79.3 | 34 | 125 | 4 |
| | | | ANTLR+Schema | 82.1 | 38 | 75 | 28 |

to generation failures or invalid query formats. We further observe that grammar filtering interacts with sampling diversity. Under higher diversity, the model generates more invalid candidates, making grammar-based filtering more effective, as it removes a larger portion of incorrect queries.

Finally, despite these improvements, some execution-time syntax errors remain. This is partly due to our use of the publicly available ANTLR-OpenCypher grammar for reproducibility, while the evaluation setup, both dataset and target databases, is based on Neo4j Cypher(v5). Since Neo4j does not support all OpenCypher constructs (e.g. GROUP BY, BETWEEN), some queries that pass grammar validation are still rejected during execution. Overall, grammar-based filtering improves syntactic validity, but introduces a trade-off between correctness and execution coverage due to more empty predictions.

**Effect of schema-aware filtering**   We next analyze the effect of schema-aware filtering on top of confidence-based and grammar-based filtering. Schema-aware filtering improves answer quality by enforcing consistency with the database schema, in particular relationship directionality. This leads to higher execution-based ROUGE-L scores across all settings, indicating more semantically correct query outputs (Table 1). The improvement is consistent (around +0.06 across settings), showing that schema constraints help select more accurate query structures. However, schema-aware filtering further reduces the number of valid candidates after grammar-based filtering. In some cases, all candidates are removed, resulting in empty outputs (Table 2). These empty predictions reduce execution coverage and lead to lower execution success rates compared to grammar-only variants.

We further observe that higher sampling diversity increases the number of structurally and semantically inconsistent queries, especially with incorrect relationship directions. In this setting, schema-based validation becomes more effective, as it filters out a larger portion of invalid candidates and improves

**Table 3**

Comparison of Gemma2 and Qwen2.5 under the very-diversed setting. Performance improvements over the base model are indicated as superscripts, scaled to percentage points for ROUGE-L scores.

| Model | Filtering | Variant | ROUGE-L (Lexical) | ROUGE-L (Exec.) | Exec. Succ. Ratio (%) |
|---|---|---|---|---|---|
| Gemma2 | Base | — | 0.6284 | 0.1680 | 63.8 |
| | Confidence | Online | $0.7099^{+8.15}$ | $0.2386^{+7.06}$ | $83.7^{+19.9}$ |
| | + Grammar | ANTLR | $0.7133^{+8.49}$ | $0.2500^{+8.20}$ | $86.7^{+22.9}$ |
| | + Schema | ANTLR + Schema | $0.6973^{+6.89}$ | $0.3025^{+13.45}$ | $85.1^{+21.3}$ |
| Qwen2.5 | Base | — | 0.6909 | 0.1922 | 68.7 |
| | Confidence | Online | $0.7135^{+2.26}$ | $0.2127^{+2.05}$ | $78.4^{+9.7}$ |
| | + Grammar | ANTLR | $0.7074^{+1.65}$ | $0.2379^{+4.57}$ | $84.5^{+15.8}$ |
| | + Schema | ANTLR + Schema | $0.6960^{+0.51}$ | $0.2534^{+6.12}$ | $82.7^{+14.0}$ |

the quality of the remaining set. Overall, schema-aware filtering improves semantic correctness of selected queries, but introduces a trade-off between answer quality and execution coverage due to more aggressive filtering. Compared to grammar-based filtering, which focuses on syntactic validity, schema-aware filtering targets semantic consistency with the database structure.

**Cross-Model Comparison** To evaluate the robustness of the proposed confidence-, grammar-, and schema-based filtering, we further test the approach on the Qwen2.5 model. Based on earlier results (Table 1), we focus on the very-diversed setting and use online confidence-based inference with the ANTLR and ANTLR+Schema variants, which showed the strongest performance. Results for both Gemma2 and Qwen2.5 are reported in Table 3.

We observe consistent trends across both models. Confidence-based filtering provides the largest improvement over the base model. Grammar-based filtering further improves structural validity, as reflected in higher execution success rates. Adding schema-aware filtering improves execution-based ROUGE-L, indicating better alignment between the execution outputs of generated queries and the ground-truth answers. On the Qwen2.5 model, syntax errors decrease substantially as filtering is applied sequentially (from 202 to 129, 64, and 68). At the same time, stricter filtering increases empty predictions (1, 0, 14, and 21), where all candidates are removed. This leads to a small drop in execution success rate and highlights the same trade-off between correctness and coverage observed in previous experiments.

# 5. Conclusion

Text2Cypher enables querying graph databases using natural language by translating user questions into executable database queries. Generated queries must satisfy both the syntax of the query language and the underlying database schema.

In this work, we study how structured constraints can be incorporated into test-time inference for Text2Cypher through a sequential filtering process. We extend a confidence-based inference framework [9] with grammar-based validation and schema-aware constraints after generation and before aggregation, allowing us to analyze how different constraint types affect query correctness. We observe that grammar-based filtering improves syntactic validity, while schema-aware filtering improves execution-based answer quality by enforcing consistency with the database structure. However, stronger filtering increases the number of empty predictions and reduces execution coverage. These findings show that simple post-generation constraints can improve the reliability of LLM-based query generation.

However, this approach has drawbacks. Confidence-based inference requires generating multiple candidates, which increases computational cost and may introduce additional latency. We leave efficiency analysis for future work. While our experiments focus on Text2Cypher, the approach is general and applicable to Text2SQL and Text2SPARQL.

## 6. Limitations

This work studies how grammar- and schema-aware constraints affect test-time inference for Text2Cypher. While the results show consistent improvements, there are several limitations. Grammar-based validation is implemented using two approaches: a simple rule-based checker and a formal ANTLR4-based parser. The rule-based method is lightweight but shallow, and may not capture complex syntactic structures. The formal grammar-based method uses the publicly available OpenCypher grammar [31], which improves correctness but depends on the completeness of the grammar. Our results also show a general trade-off of stronger constraints. While grammar-based validation improves syntactic validity and schema-aware constraints improve answer quality, both can increase the number of empty predictions and reduce execution coverage. Finally, the added filtering steps may increase inference latency. We have not studied efficiency or runtime cost in detail and leave this for future work. Although we focus on Text2Cypher, the approach can be applied to other structured generation tasks such as Text2SQL and Text2SPARQL.

## Declaration on Generative AI

During the preparation of this work, the author(s) used ChatGPT in order to: Grammar and spelling check, Paraphrase and reword, Improve writing style. After using these tool(s)/service(s), the author(s) reviewed and edited the content as needed and take(s) full responsibility for the publication's content.

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
