# OpenReview forum: "Extending Confidence-Based Text2Cypher with Grammar and Schema Aware Filtering"
_ijcai.org/IJCAI-ECAI/2026/Workshop/GENAIK-NORA — IJCAI-ECAI 2026 Joint Workshop on GENAIK and NORA_

### Official Review · Reviewer_buxZ · 2026-06-05
**Important Problem and Sound Experimental Design**

**Rating:** 6
**Confidence:** 3

**Review:**

This paper presents a study on improving the correctness of LLM-generated Cypher queries at test time. The authors extend a confidence-based inference framework by introducing a sequential post-generation filtering pipeline. This pipeline first filters low-confidence query candidates, then applies grammar-based validation (using both a naive and a formal ANTLR-based approach) to ensure syntactic correctness, and finally uses schema-aware filtering (specifically checking relationship directionality) to enforce semantic consistency with the database. The methodology is evaluated on a challenging subset of the Text2Cypher dataset using two different instruction-tuned LLMs (Gemma2 and Qwen2.5). The key findings highlight that while grammar and schema filtering significantly improve syntactic validity and execution-based answer quality, respectively, they come at the cost of reduced execution coverage due to an increase in empty predictions where all candidates are filtered out.

This is a solid, well-executed piece of research that provides valuable insights into test-time strategies for improving Text2Cypher generation. The experimental analysis is thorough and the conclusions are well-supported by the data. While the paper could be strengthened by a more direct comparison with decoding-time methods and a deeper analysis of the failure cases, its current contributions are sufficient for acceptance. The work is a valuable addition to the literature on building more reliable natural language interfaces for databases.

---

### Official Review · Reviewer_7DCA · 2026-06-05
**Incremental contributions compared to prior work**

**Rating:** 5
**Confidence:** 4

**Review:**

The paper outlines how structured constraints can be used to improve Test2Cypher pipelines during test-time inference. The paper introduces two extra constrains that are added on top of prior work as part of a workshop paper, which introduced confidence based filtering for Text2Chypher. The author adds grammar validation(rule-based or formal approaches like ANTLR4) and schema constraints(relational direction) as additional filtering stages, on top of the previously described pipeline for Text2Cypher from the prior work.  Experiments are conducted on the same 789 sample set described in the prior work, with the same base model and inference setup.

# Strengths
- Well structured and well written
- Ablation study shows value of each of the filtering techniques
- Trade-offs are discussed without overrepresentation of gains, with description on added inference latency, which is described as future work
- Experiments are meaningful, results are promising
# Weaknesses
- Very incremental contributions compared to the authors' prior work paper. The prior work already describes the pipeline, setup, data, confidence based modules, as well as the evaluation framework.
- Main contributions of this paper relative to the prior work is only two filtering components, which is a pretty narrow scope for a standalone paper
- Paper would benefit from a longer discussion of mismatch in the grammar filtering stage. The OpenCypher grammar and Neo4j Cypher diverge on constructs, which the authors acknowledge, but it might actually undermine the results at this stage since there may be noise. The invalid ones would need a systematic analysis to ensure the numbers for this stage are meaningful
- Although the paper describes as introducing "schema aware" filtering, it is mainly focusing on relationship direction, which is quite simple
- Since the contributions are incremental, fallbacks could have been discussed on failure cases

---

### Official Review · Reviewer_Mpf3 · 2026-06-06
**Paper discussing its own limitations**

**Rating:** 5
**Confidence:** 3

**Review:**

I consider the paper questionable in its logical structure and consequently in its overall contribution.
Detecting incorrect formal language queries, i.e. those violating grammar constraints or database semantics, can be considered a near-perfect task. Filtering incorrect queries from a set of automatically generated ones improves set quality _by_construction_ under any metric, for any task. Such improvement can be just assumed, and any effort in showing it brings poor information. That is a first general objection. In addition, the paper faces its particular own implementation issues as 1) noisy, imperfect filtering, and 2) scarcity of generated queries, which may result in an empty set upon filtering, and worsening results. So, the paper focuses on measuring and discussing the effects of such two self-introduced limitations, arguing that a tradeoff should be pursued in the application of filters. Instead, I object that straightforward application of formal filtering should not be under discussion, that any effort should be directed to fixing the problems (which are totally unrelated to the task), and that the reader has no benefit from current temporary results while he is waiting for those fixes. All that being said, I acknowledge that the spirit of a workshop may also promote plain reporting of current experience and work in progress, as the present one.

---

### Official Review · Reviewer_qCXy · 2026-06-06
**Post generation filtering for improving the results**

**Rating:** 5
**Confidence:** 4

**Review:**

This work examines how to improve the reliability of Text2Cypher generation by incorporating structured, post‑generation constraints during test‑time inference. Although large language models can translate natural‑language questions into database queries, executable outputs must satisfy both syntactic rules and schema constraints—requirements that are typically addressed through fine‑tuning or prompt‑based constraint injection. The authors extend a confidence‑based inference framework with a sequential filtering pipeline that applies confidence scoring, grammar validation, and schema‑aware consistency checks before aggregating candidate queries. Experiments show that grammar‑based filtering improves syntactic validity, while schema‑aware filtering further enhances execution correctness by enforcing alignment with the underlying database structure. However, the study does not analyze computational efficiency, an essential factor for real‑world deployment. Without runtime and cost evaluation, it is difficult to fully assess the practical value of the proposed approach.

---

### Decision · Program_Chairs · 2026-06-10

Accept